# Theory-based immunisation health education intervention in improving child immunisation uptake among antenatal mothers attending federal medical centre in Nigeria: A study protocol for a randomized controlled trial

**Abubakar Nasiru Galadima**[1], **Nor Afiah Mohd Zulkefli**[1] *, **Salmiah Md Said**[1], **Norliza Ahmad**[1], **Saleh Ngaski Garba**[2]

1 Department of Community Health, Faculty of Medicine and Health Sciences, University Putra Malaysia, Seri Kembangan, Malaysia, 2 Department of Nursing Sciences, Faculty of Allied Health Sciences, Ahmad Bello University, Zaria, Nigeria

* norafiah@upm.edu.my

**Data Availability Statement:** No datasets were generated or analysed during the current study. All

## Abstract

### Background

Childhood immunisation coverage is very low in Nigeria (31%) with Zamfara State being amongst the states with the poorest coverage (<10%). Lack of maternal knowledge, attitude, outcome expectations, self-efficacy, cultural beliefs and assumptions of religious regulations of antenatal mothers towards childhood immunisation are the contributory factors to poor childhood immunisation uptake. This study aims is to develop, implement and evaluate the effects of an immunisation health educational intervention with application of Social Cognitive Theory on pregnant women to improve knowledge, attitude, outcome expectations, self-efficacy, cultural beliefs and assumptions on religious regulations regarding childhood immunization uptake in Federal Medical Centre Gusau, Zamfara State, Nigeria.

### Methodology

The study will be a single-blind parallel-group randomised controlled trial, where baseline data will be collected from 392 estimated antenatal mothers, after that they will be evenly randomised using randomly generated permuted block sizes (each containing two intervention and two control assignments). The study participants will be antenatal mothers of ages 18 years and above who are in third trimesters and attending Federal Medical Centre Gusau, Zamfara State, Nigeria; during the study period and fulfilled all the inclusion and exclusion criteria. The intervention group will undergo five-health education sessions on immunisation, which will be strictly guided by Social Cognitive Theory-based intervention module: while the control group will receive usual care (standard care). Follow-up data will be collected using the same questionnaire at 6-weeks post-delivery, 10-weeks post-delivery and 14-weeks post-delivery. The generalized linear mixed model will be carried-out to determine the overall effect of the intervention after controlling for 14 potential confounding

relevant data from this study will be made available upon study completion.

**Funding:** The author(s) received no specific funding for this work.

**Competing interests:** The authors have declared that no competing interests exist.

**Abbreviations:** VPDs, Vaccine Preventable Diseases; EPI, Expanded Programme on Immunization; SCT, Social Cognitive Theory.

variables. An intention to treat analysis will also be carried-out. Childhood immunisation uptake is the primary outcome while the secondary outcomes are: improved knowledge scores, attitude scores, outcomes expectation, self-efficacy scores, cultural beliefs scores and assumptions on religious regulations scores.

## Discussion

The study will be a randomised controlled trial, that focuses on the effects of an immunisation health educational intervention with application of Social Cognitive Theory on pregnant women to improve knowledge, attitude, outcome expectations, self-efficacy, cultural beliefs and assumptions on religious regulations regarding childhood immunisation uptake in Federal Medical Centre Gusau, Zamfara State, Nigeria.

## Trial registration

Pan African Clinical Trial Registry PACTR202006722055635. Protocol registered on 09 June 2020.

## Background

The immune system of children is stimulated by vaccines to fight against diseases or infections. It is a well-established fact that immunisation is a tool that can effectively control and eradicate life-threatening infectious diseases [1]. Immunisation is considered as one of the most successful and cost-effective public health sustainable interventions for human beings against diseases that affect our health [2]. Routine immunisation plays an important role in reduction of child mortality, resulting from vaccine preventable diseases (VPDs). WHO revealed that immunisation has been estimated to prevent 3 million deaths globally every year [3].

Despite the efforts made by health organizations to combat diseases, vaccine preventable diseases have remained the most common cause of childhood mortality, which is estimated at 3 million deaths around the world annually [4]. In Nigeria, VPDs accounted for 17% and 22% of under-five childhood morbidity and mortality respectively [5].

In Nigeria, routine vaccination coverage for all recommended vaccines has remained poor. However, there was an increase in vaccination coverage from 25.4% of eligible children (12–23 months of age) in 2013 to 31% in 2018 [6]. In Zamfara State, only 7.4% of the children are fully immunized [6]. Table 1 shows the individual vaccine uptakes comparing Zamfara state with Nigeria. According to the Expand Program on Immunisation (EPI), every child in Africa must receive one dose of Bacillus Calmette Guerin (BCG), Oral Polio Vaccine (OPV0) and Hepatitis B Vaccine (HBV1) at birth, Penta1 & OPV1 at 6 weeks of age, Penta2 & OPV2 at 10 weeks of age, Penta3 & OPV3 at 14 weeks old and measles and yellow fever at 9 months old [6].

The World Health Organization (WHO) uses diphtheria, pertussis and tetanus (DPT3) coverage to monitor immunisation system performances across the districts of a country or territory. It indicates that a child has access to immunisation services and that the services were utilised; that is, they returned on at least 3 occasions [7]. The coverage for DPT3 vaccine varies across the states, for-instance, in Zamfara State only 10.8% of the children received the DPT3 vaccine, whereas at national level 50.1% of the children received the DPT3 vaccine [6]. These coverages are still far below the targets endorsed by WHO in 2012 Global Vaccine Action Plan, which aimed at ensuring delivery of universal access to immunisation with associated

**Table 1. Individual vaccine uptakes.**

| Vaccines | BCG | DPT1 | DPT3 | OPV3 | Measles |
|---|---|---|---|---|---|
| Nigeria | 66.7% | 65.3% | 50.1% | 47.2% | 54% |
| Zamfara State | 16.1% | 17.1% | 10.8% | 15.2% | 12.2% |

Nigeria Demographic and Health Survey 2018 [6]

targets of reaching 90% national vaccination coverage and at least 80% vaccination coverage in every district [8].

Many observational studies conducted in Nigeria and other countries that focused on examining the factors influencing childhood immunisation uptake has discovered some personal factors: parental sociodemographic factors [4, 9–30], obstetric factors [10, 12, 15, 18, 25–29, 31–33], maternal knowledge gaps [10, 12, 13, 16, 20, 23, 25, 31, 34–38], maternal attitudes and self-efficacy [12–14, 36, 39–43], maternal outcome expectations [14, 35, 44–46]; environmental factors: physical and social factors influencing childhood immunisation uptake [18, 20, 22, 41, 47–52].

Despite the consistent poor childhood immunisation uptake, inadequate knowledge of mothers towards childhood immunisation is another considerable factor that has over the years contributed to poor attitude and practices of mothers towards childhood immunisation uptake among Nigerian mothers. Based on the literature search, only two health education intervention studies were conducted on increasing childhood immunisation uptake and both used a community based randomised-controlled trial. The first study was a controlled community trial conducted in Sokoto State, Nigeria which focused on evaluating the impact of immunisation health education intervention using Community Level Nutrition Information System for Action (COLINSA) strategy on knowledge and practice of childhood immunisation among mothers'.' At the end of the study, the proportion of children in the intervention group who received DPT1 was 35% and DPT3 33% while in the control group DPT1 was 25% and DPT3 was 20%. However, this finding is not statistically significant p = 0.734 [53].

The second study which was a randomised controlled trial, was carried-out to evaluate the effect of a low-literacy immunisation promotion educational intervention in improving DPT/Hepatitis B vaccine uptakes among 366 mother-infant pairs with infants aged ≤6 weeks of age in Karachi, Pakistan. At four months after enrolment, among 179 mother-infant pairs in the intervention group, 129 (72.1%) had received all the 3 doses of DPT/Hepatitis B vaccine while in the control group only 92/178 (51.7%) received all the 3 doses. This finding was statistically significant (ARR = 1.39; 95% CI: 1.06–1.81) [54]. However, none of the studies was guided by any health theory in providing a strategy and a step-by-step summary of specific factors to be considered when developing, implementing, and assessing the health education interventions. Notwithstanding the usual health education given to pregnant women attending antenatal at FMC Gusau, child immunisation uptake is still low. It is important therefore, to compare the results of our intervention group and that of control group to help in evaluating the effectiveness of the intervention.

Therefore, there is a need to develop and implement a theory-based intervention with application of social cognitive theory to evaluate the effectiveness of immunisation health education in improving the knowledge, attitude, outcome expectation, self-efficacy; cultural beliefs and assumptions on religious regulations of antenatal mothers which will hopefully increase childhood immunisation uptake and decrease mortality and morbidity among children in Nigeria. The intervention module that will be developed from this study may be adopted and consolidated into the routine health education delivering to pregnant women at the time of their

antenatal care visits. This will provide better knowledge to antenatal mothers concerning the importance of child immunisation. The study findings could benefit policy makers in making decisions and formulating appropriate guidelines and policies with regards to childhood immunisation uptake in order to reduce childhood mortality and morbidity of VPDs. Other researchers may also adopt the validated questionnaire and intervention modules that will be developed from this research for future studies.

The research questions of this study are as follows:

1. What is the effect of an immunisation health educational intervention on the knowledge, attitude, outcome expectation, self-efficacy, cultural beliefs and assumptions of religious regulations among pregnant women attending Federal Medical Centre Gusau, Zamfara State, Nigeria?

2. What is the effect of an immunisation health educational intervention on childhood immunization uptake in Federal Medical Centre Gusau, Zamfara State, Nigeria?

The proposed study will be single-blind parallel-group randomised controlled trial and will be guided using CONSORT Statement [55]. The participants will be randomised into either the intervention group which will received the SCT based health educational intervention on childhood immunisation, or the control group which will receive usual care (standard care).

## Trial aims and objectives

The aim of this randomized controlled trials (RCT) is to develop, implement and evaluate the effects of an immunisation health educational intervention with application of Social Cognitive Theory on pregnant women to improve childhood immunisation uptake among children in Federal Medical Centre Gusau, Zamfara State, Nigeria.

## Specific objectives

The specific objectives of this study are:

1. To determine and compare the baseline of respondents between intervention and control group at baseline according to:

   a. Parental socio-demographic characteristics (maternal age, maternal education, paternal education, maternal marital status, maternal occupation, area of residence, number of siblings, religion, ethnicity and family income).

   b. Obstetric history (antenatal care follow-up, Maternal Tetanus Toxoid (MTT) vaccine and preceding birth intervals).

   c. Knowledge scores on (immunization, knowledge on VPDs and knowledge on EPI schedule).

   d. Outcome expectation scores (benefit of immunization and seriousness of VPDs).

   e. Attitudes scores (maternal perceptions towards childhood immunization).

   f. Self-efficacy scores (vaccine safety and vaccine side effect).

   g. Environmental factors (assumptions of religious regulations scores, cultural belief scores, distance to health facility, mode of transportation, hospital accessibility and attitude of hospital staff).

2. To develop and implement an immunization health educational intervention based on Social Cognitive Theory for pregnant women in Federal Medical Centre Gusau, Zamfara State, Nigeria.

3. To compare the total immunization knowledge, attitude, outcome expectation, self-efficacy, cultural beliefs and assumptions of religious regulations scores within the intervention and control group at baseline to 6-weeks post-delivery, 10-weeks post-delivery and 14-weeks post-delivery.

4. To compare the knowledge, attitude, outcome expectation, self-efficacy, cultural beliefs and assumptions of religious regulations scores between the intervention and control group at baseline to 6-weeks post-delivery, 10-weeks post-delivery and 14-weeks post-delivery.

5. To determine the effects of the intervention on improving child immunization uptake in Federal Medical Centre Gusau, Zamfara State, Nigeria.

## Methods

The study protocol was developed in accordance with Standard Items Recommendations for Interventional Trials (SPIRIT) guidelines [56] and it has been registered on 09 June 2020 with Pan African Clinical Trial Registry PACTR202006722055635 *(Refer to S1 Checklist)*.

### Study design

The study will be single-blind parallel-group randomized controlled trial. The participants will be randomised into either the intervention group which will received the SCT based health educational intervention on childhood immunisation, or the control group which will receive usual care (standard care).

### Study setting and sampling population

The study setting will be Federal Medical Centre, Gusau Zamfara State, Nigeria. The sampling population of this proposed study will be pregnant women who will attend the hospital during the recruitment period. The study participants will be selected to participate according to pre-defined inclusion and exclusion criteria of the protocol.

### Eligibility criteria

The eligibility criteria for inclusion into the study will be as follows:

Antenatal mothers of age 18 years and above who are in third trimesters (28–35 gestational weeks); mothers with single pregnancy; must be mothers who planned to deliver and intended to attend the clinic to immunise their children at Federal Medical Centre Gusau; citizens of Nigeria and those resident in Gusau.

The following will be excluded from the study:

Mothers without good understanding of Hausa language as the intervention will be delivered in Hausa language; mothers who attended previous health education intervention on immunization will be excluded to minimise confounding; mothers living in the same household to minimise contamination; mothers of family with nomadic lifestyle will also be excluded as previous study revealed low immunisation uptake among children whose mothers' practice nomadic lifestyle [20].

### Randomisation

Random assignment of the participants will be done on the same day that the participants will be selected into the study; the participants will be randomised to either the intervention (Immunization Health Education Intervention) or control group (usual care). However, it will be carried out before taking the baseline date.

### Sample size determination

The minimum sample size required to test the proposed study hypotheses has been determined using the formula for comparing two proportions in selected samples of both intervention and control groups [57]. This formula gave the required number (n) for each group. This provided a total sample size of 196 participants in each group, which will then be multiplied by two since the study is comprised of two arms. Therefore, 392 are the calculated sample size required to test the hypothesis for this study *(Refer to S1 File)*.

### Random selection

In this study, the participants will be recruited within the eight succeeding booking sessions, which will hopefully provide a total sampling frame of 800 clients. The study participants will be selected using systematic random sampling technique from the sampling frame; K$^{th}$ element of two will also be applied (800 divided by the sample size of 392). One of the first two eligible clients on the list will be selected using the table of random numbers. Afterwards, the selection of the second client will be made serially using the list until the last eligible client is selected. Based on the same sequence in the sampling frame, the list of participants will then be organized, with serial numbers assigned to them for the purpose of identification. Subsequently, the use of these serial numbers will be employed in random allocation for both groups. The sampling frame list will be prepared and provided by the hospital staff working in the Medical Records Department. Likewise, the recruitment based on the table random numbers will be performed by the hospital nurse from Antenatal Care Department. Both of them will not be participated in any of the study process.

### Sequence generation

In the current study, sequence generation will be performed by the hospital staff working in the Medical Records Department. The staff will be trained to perform the sequence generation. However, these staff will not be part of any process for this proposed research. A permuted block randomization technique, having equal size permuted blocks of four (each containing two for interventions and two for control group) will be used. Since the calculated sample size for this study is three hundred and nighty two, 98 permuted blocks will be required. Nevertheless, in order to avoid the problem of under-sampling, a little higher than the calculated sample size will be targeted for recruitment [58]. Thus, 110 permuted blocks containing all the possible combinations (for example; BABA, ABBA, AABB, ABAB, BBAA, BAAB) will be generated using the random function in Microsoft Excel.

### Allocation concealment mechanism

Opaque envelopes will be used to place in the sequences that will be generated and the medical records' staff that will generate the sequence will then seal them. In order to guide the people responsible for the allocation the envelopes will also be serially numbered from the outside.

## Implementation

Two staff of the antenatal care unit will assist in assigning groups to the participants independently. The first staff will hand over the sealed envelopes to the participants in a serial order, and afterwards, the staff will be asked to match the serial numbers on the allocation envelopes with the participants' given serial numbers succeeding the random sampling. Subsequently, the participants will be directed to the second staff, who will then open the envelopes and inform them of the date to come for their health education session based on the group they fall under. After this is done, the group to which the participant belong to, will be documented on a sheet provided. All the study participants will be given a hand card, which is the study pro forma containing the child's serial number and bio-data. Also, the hand card will contain a short note that shows they have been enrolled into a follow-up study, and seeking any of the attending medical staff (nurse) to kindly assist in completing the post-natal findings as indicated on the card.

## Blinding

The proposed study will be single-blinded; the study enumerators will be blinded against the participants' groups. In any of the processes either the group allocation, or delivering of the intervention, the enumerators will not be involved, and the participants' groups will also not to be specified on their cards. The staff that will document the list of the assignment will keep it confidential, the staff will also be aware of the group coding (A and B), however, the staff will be unaware of the interventions that will be assigned to them.

## Intervention

The participants in the intervention group will receive the "Immunisation Health Education Intervention" while the participants in the control group will receive "usual care". The proposed interventions will be implemented at 35 gestational weeks after the baseline data has been collected from the participants. Based on the recommendation of McMillan [59], educational interventions should be carried out using approaches that are in accordance with what is used in practice. It is based on this, that the delivery of educational interventions in this study will be done at group levels in a two session. Every session will be composed of forty participants, while the facilitator of each of the sessions will be a midwife nurse.

The process of module delivery will be closely monitored and supervised by the investigator, and the relevant feed-back will be given to the facilitator in order to ensure that the module is followed accordingly. The five modules will be delivered by the facilitator, who will ensure that the participants are actively engaged. It is estimated that the delivery sessions will last for 30 minutes, 30 minutes, 30 minutes, 1 hour, 30 minutes, 45 minutes, 1 hour and 30 minutes. For modules 1, 2, 3, 4 and 5 respectively *(Refer to S2 File)*. The participants will be allowed to take short breaks of about 10 to 15 minutes between modules. Each participant will be given the chance to demonstrate what she has learnt, and corrective feed-back will be provided at the end of the modules. A compensation fees of (500 Naira) approximately 1.2 USD will be given to the participants in the intervention group in favour of their transportation. The mothers in the intervention group will also be receiving a mobile call reminder at 7 days and 1 day before their childhood immunisation schedule. The participants in the control group will not receive any mobile call reminder. Any mother in either the intervention or control group who completed her child vaccination from 0-doses to penta-3 will receive a baby pampers at the end of the intervention which will cost N700 (Naira) approximately 1.5 USD as recommended by Bandura's [60].

The usual care (standard care) given to antenatal mothers attending FMC Gusau will comprised of general lectures regarding birth preparedness, danger signs in pregnancy, malaria

complications, preventions and treatments, breast feeding, postnatal check-up, immunisations, family planning and interactive sessions. These general lectures are always given to antenatal mothers during every visit which last for about two hours.

## Outcomes

The primary outcome for the proposed study is childhood immunisation uptake and the secondary outcomes are: improved knowledge scores, improved attitude scores, improved outcomes expectation scores, improved self-efficacy scores, improved cultural beliefs scores and improved assumptions of religious regulations scores. They will be taken at 6 weeks post-delivery, 10 weeks post-delivery and 14 weeks post-delivery (*Refer to S3 File*).

## Participants timeline

The recruitment time will be seven weeks and the study participants will be recruited at their 24 to 33 weeks period of amenorrhea. After completing the intervention, the study participants will be followed-up at three times points (6-weeks post-delivery, 10-weeks post-delivery and 14-weeks post-delivery). The expected duration of participation in the study is 22 weeks approximately 6 months.

## Ethics approval and consent to participate

Ethical approval for the proposed study was received from both the Jawatankuasa Etika Universiti Putra Malaysia (JKEUPM) (reference number: JKEUPM-2020-398) and Federal Medical Centre Gusau, Zamfara State (reference number: FMC/2019/985/23). A respondent's information sheet in Hausa language will be provided by a nurse from antenatal care unit to each participant, from whom informed consent will be obtained. The study has been registered with the "Pan African Clinical Trials Registry" with unique identification number for the registry PACTR202006722055635. The participants information that will be provided and their identity will remain confidential and solely for the purpose of this research. Participation in the study will be voluntary and any of the study participants also has the authority to withdraw at any point of the intervention (*Refer to S4 File*).

## Study process

An illustration of the study process was done in accordance with CONSORT statement [55]. During the eight weeks recruitment period, a total of 551 pregnant women had been assessed for possible inclusion in the study out of which 159 did not meet the inclusion criteria while ten declined to participate. The remaining three hundred and eighty-two participants were then randomly assigned equally, to either the intervention or the control group. SPIRIT schedule of the study and flow diagram of the study are shown in Figs 1 and 2 respectively. The baseline data of the study participants has been taken and they will both be followed up at 6 weeks post-delivery, 10 weeks post-delivery and 14 weeks post-delivery to assess the effect of the intervention regarding their knowledge, attitude, outcome expectations, self-efficacy, cultural beliefs and assumptions of religious regulations towards child immunization, as well as to assess their child immunization uptake.

## Study instrument

The instruments that will be used in this proposed study includes the immunisation health educational intervention modules and the data collection tools; which will comprise of a questionnaire and the study pro forma will also be used.

Template of recommended content for the schedule of enrolment, interventions, and assessments.

| | STUDY PERIOD | | | | |
| --- | --- | --- | --- | --- | --- |
| | Enrolment | Allocation | Post-allocation | | |
| TIMEPOINT** | $-t_1$ | 0 | $t_1$ | $t_2$ | $t_3$ |
| **ENROLMENT:** | | | | | |
| **Eligibility screen** | X | | | | |
| **Informed consent** | X | | | | |
| **Allocation** | | X | | | |
| **INTERVENTION GROUP** | | X | | | |
| **CONTROL GROUP** | | X | | | |
| **ASSESSMENTS:** | | | | | |
| *Knowledge Scores* | X | | | | |
| **Attitude Scores** | X | | | | |
| **Outcome Expectations Scores** | X | | | | |
| **Cultural Belief Scores** | X | | | | |
| **Assumptions on Religious Regulations Scores** | X | | | | |
| **Self-efficacy Scores** | X | | | | |

**Fig 1. SPIRIT schedule of enrolment.**

### The immunisation health education intervention modules

**Development of the immunisation health education intervention modules.** The sources of information that will be used in developing the modules include:

I. Basic Guidelines for Routine Immunisation Service Provider (Nigeria Federal Ministry of Health).

II. World Health Organization publications on immunisation.

III. Publications from studies conducted in Nigeria on childhood immunisation.

The study process will be illustrated according to the CONSORT statement [56].

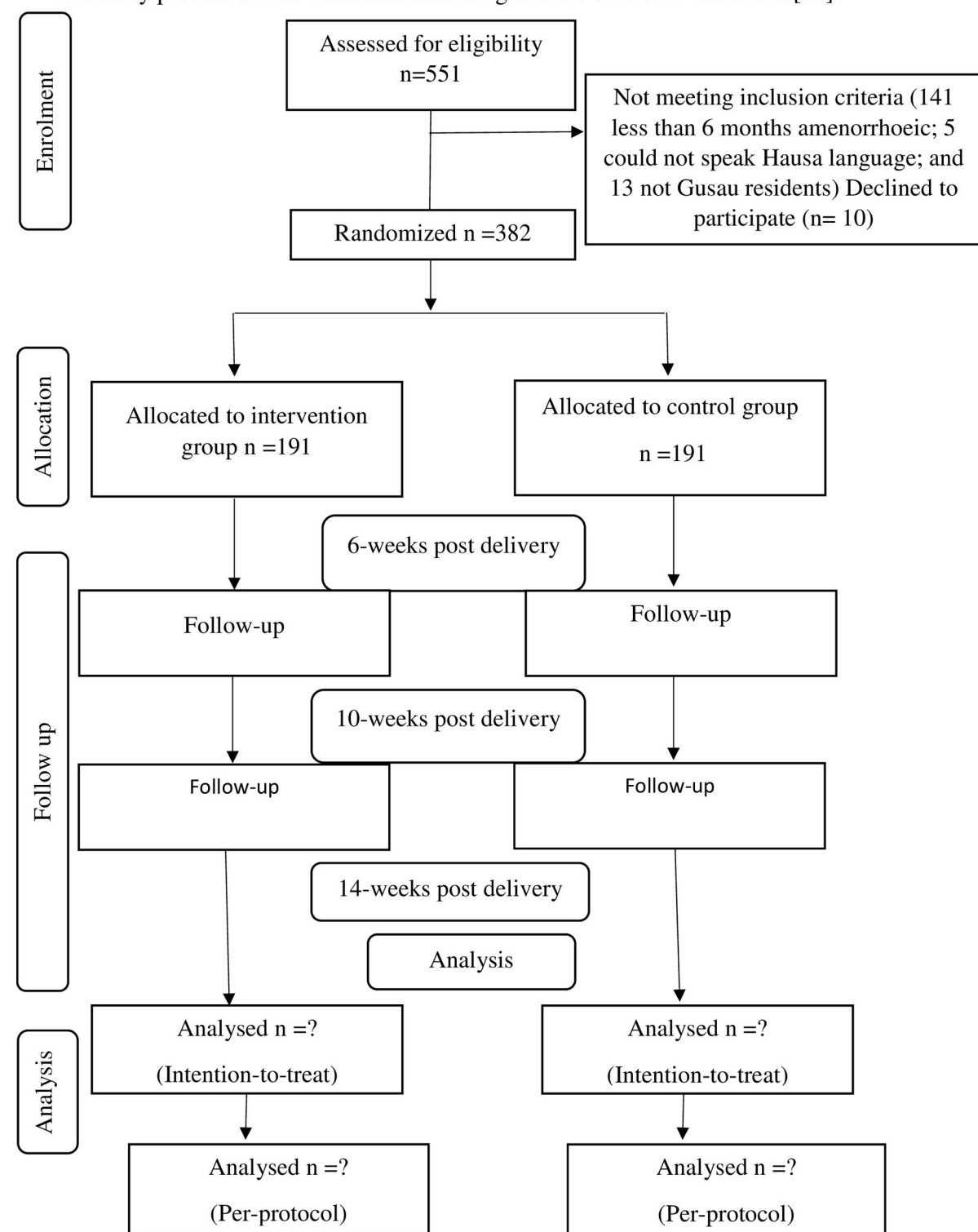

**Fig 2. Consort study flow chart of the intervention and control groups.**

The modules were developed based on the constructs of social cognitive theory.

**Structure of the immunisation health educational intervention modules.** The intervention will be comprised of five modules, designed to cover each of the constructs of the SCT as presented in S2 File. The knowledge construct will be covered by the first and second modules, the third module will cover the attitude construct, the fourth module will cover the outcome expectation construct, while the fifth module will cover the construct for self-efficacy *(Refer to S2 File)*.

**Training of the module facilitator.** The modules will be delivered by a midwife, who will also serve as a facilitator at all the health education sessions during antenatal visits. The researcher shall deliver a one-time one-on-one training session to the facilitator. The training session, which will involve teaching the midwife how to deliver the contents to the respondents, will last for about 4 hours. Before the midwives start delivering the modules, another session will be held to revise what they were taught in the previous session, so as to ensure that they are properly equipped to deliver the modules.

**Quality control of the module contents.** The modules for the intervention group were appraised for their contents and relevance by expert teams. For the immunisation health educational intervention modules, the team comprised of four public health specialists and a medical sociologist, a health educator, and a mid-wife. The intervention will be pre-tested with a sample of twenty pregnant women (35 gestational weeks) from the same hospital, who will not be part of the study. Appropriate feedback will then be obtained from the participants; this will include knowledge, attitude, outcome expectation, self-efficacy, cultural beliefs and assumptions of religious regulations components and appropriate modifications and corrections will be further effected.

## Data collection methods

The data collection tools that will be used comprised of a questionnaire and the study pro forma *(Refer to S5 and S6 Files)*.

## Questionnaire

**Questionnaire development.** The study will use a guided questionnaire in collecting the necessary information. The questionnaire is first developed in English language. Its content validity has been assessed by an expert group [61], which involved six public health specialists and one medical sociologist who went through the questionnaire to ensure that the wordings of its items are clear, and that they represent their content domain [62]. The questionnaire was later translated into Hausa language via the process of translation and adaptation of instruments figured by the World Health Organization [63].

**Questionnaire quality control.** This will be entailed ensuring the validity and reliability of the questionnaire, as well as adequately training the enumerators. The questionnaire Content Validity Ratio (CVR) was carried-out while face validity and exploratory factor analysis will be performed before the commencement of the intervention.

**Study pro forma.** The researcher designed a study hand cards which will be used to collect information about childhood immunization uptake. The hand cards will be given immediately after recruitment, and will be submitted to the researcher at three months post-delivery after Penta 3 uptake. The card has participant's serial number, basic bio-data, and a short appeal to the attending physician or nurse, informing them that the patient is enrolling into a follow-up study, and his/her help is needed to kindly fill the blanks provided on the card. The items it contained are: childhood immunization uptake and date given (BCG, Hep B-0 and OPV-0; OPV-1, PCV-1 and Penta-1; OPV-2, PCV-2 and Penta-2; OPV-3, PCV-3 and Penta-3).

**Data analyses.** The data analyses will be conducted using IBM Statistical Package for Social Sciences (SPSS) version 25. The data that will be collected from questionnaire and the child immunization record will be entered into the software, followed by data cleaning.

Maternal age, knowledge scores, attitude scores, outcome expectation scores, self-efficacy scores, cultural beliefs scores and assumptions of religious regulations scores will be subjected to both statistical and graphical normality tests due to the fact that they are continuous variables.

Mean and standard deviation will be used as the measures of central tendency and dispersion to summarise continuous variables that are normally distributed while mean or median will be used to summarised skewed data. Categorical data will be summarised as frequency and percentage.

Chi-squared test will be carried-out to make a baseline comparison of the groups by their sociodemographic characteristics, obstetrics factors, healthcare system factors, responses on immunisation knowledge, attitude, outcome expectation, self-efficacy cultural beliefs and assumptions on religious regulations. Fisher's Exact test will be conducted for variables where up to 20% of the cells have less than five observations. Independent t-test will be carryout to determine the between-group differences in the knowledge scores, attitude scores, outcome expectation scores, self-efficacy scores, cultural beliefs scores and assumptions of religious regulations scores at three time points.

In order to determine within-group differences with time for knowledge scores, attitude scores, outcome expectation scores, self-efficacy scores, cultural beliefs scores and assumptions of religious regulations scores the researcher will conduct one-way repeated measures analysis of variance (ANOVA) for these variables if there is no substantial non-normality of the residuals. Intention-to-treat analysis (ITT) will be carried-out separately for each group and replacement of missing data will be done through using multiple imputations technique. The estimated outputs that will be obtained after ITT analysis will then be pooled into a single data set, after that, the Generalized Linear Mixed Models (GLMM) analysis will be carried-out to investigate the overall effectiveness of the intervention using 95% Confidence Interval (CI) and 0.05 will be set as the level of significance. Two measures of effect size will be used in this study: Partial Eta squared and the coefficients of the fixed effects. Partial Eta squared will be used in the mixed design repeated measures ANOVA, and will be classified as small, medium or large effect (0.2, 0.5 and 0.8) respectively according to the magnitude. The coefficients of the fixed effects will be used in the GLMM analysis.

In order to check the robustness of the result, sensitivity analysis will be performed to determine the effect of drop out and non-response on the findings of the proposed study, the GLMM analysis will be repeated again without replacing the missing values. The differences between the coefficients obtained after replacement (intention to treat analysis) and before replacement of the missing values (per-protocol) will then be calculated and expressed as percentages of the former.

## Discussion

To our knowledge, this will be the first immunisation health education intervention (RCT) to be conducted with application of health theory targeting antenatal mothers. The first study was a controlled community trial conducted in Sokoto State, Nigeria which focused on evaluating the impact of immunisation health education intervention using Community Level Nutrition Information System for Action (COLINSA) strategy on knowledge and practice of childhood immunization among mothers. At the end of the study, the proportion of children in the intervention group who received DPT1 was 35% and DPT3 33% while in the control group DPT1 was 25% and DPT3 was 20% [53]. The second study which was a randomised

controlled trial, was carried-out to evaluate the effect of a low-literacy immunisation promotion educational intervention in improving DPT/Hepatitis B vaccine uptakes among 366 mother-infant pairs with infants aged ≤6 weeks of age in Karachi, Pakistan. At four months after enrolment, among 179 mother-infant pairs in the intervention group, 129 (72.1%) had received all 3 doses of DPT/Hepatitis B vaccine while in the control group only 92/178 (51.7%) received the all 3 doses [54]. Social cognitive theory (SCT) is one of the health theories commonly used in health education interventions [64, 65] which describes human behaviour through the influence of personal and environmental factors [60]. The theory accounts for human behaviour, cognition and environment and is the only health theory that takes into account reciprocal interaction, unlike other theories such as Information Motivation Behavioural Skills model [64, 66]. Many personal and environmental factors have been found to influence childhood immunisation uptake in African countries including Nigeria. Therefore, using SCT may enable researchers to address both personal and environmental factors influencing childhood immunisation uptake. Our study will ensure maternal knowledge, attitude, outcome expectations, self-efficacy, cultural beliefs and assumptions of religious regulations regarding childhood immunization uptake are increased at the end of the intervention. The study findings will be communicated to participants, healthcare professionals, the public, and other relevant groups via publication.

## Protocol amendments

Any important protocol modifications for-instance, changes in the eligibility criteria, outcomes or analysis will be communicated to Pan African Clinical Trials Registry and Jawatankuasa Etika Universiti Putra Malaysia.

## Supporting information

**S1 Checklist. SPIRIT 2013 checklist: Recommended items to address in a clinical trial protocol and related documents**[*]**.**
(DOC)

**S1 File. Sample size determination.**
(DOCX)

**S2 File. Component of the intervention.**
(DOCX)

**S3 File. Operational definitions.**
(DOCX)

**S4 File. Participants consent form.**
(DOC)

**S5 File. Questionnaire.**
(DOCX)

**S6 File. Study pro forma.**
(DOCX)

## Acknowledgments

The authors will acknowledge and wish to express their appreciations to the women who will participate in this study. They will also thank the Emir of Gusau Dr Ibrahim Bello (traditional

leader) and Sheikh Ahmad Umar Kanoma (Religious leader), enumerators, all ante-natal care and immunization unit staff of the Federal Medical Centre, Gusau, Zamfara state Nigeria for their tremendous support that will be received.

## Author Contributions

**Conceptualization:** Abubakar Nasiru Galadima, Nor Afiah Mohd Zulkefli, Salmiah Md Said, Norliza Ahmad.

**Data curation:** Abubakar Nasiru Galadima, Nor Afiah Mohd Zulkefli.

**Formal analysis:** Abubakar Nasiru Galadima, Salmiah Md Said.

**Investigation:** Abubakar Nasiru Galadima.

**Methodology:** Abubakar Nasiru Galadima, Nor Afiah Mohd Zulkefli, Salmiah Md Said, Norliza Ahmad, Saleh Ngaski Garba.

**Software:** Abubakar Nasiru Galadima.

**Supervision:** Abubakar Nasiru Galadima, Nor Afiah Mohd Zulkefli.

**Validation:** Abubakar Nasiru Galadima, Nor Afiah Mohd Zulkefli, Salmiah Md Said, Norliza Ahmad, Saleh Ngaski Garba.

**Visualization:** Abubakar Nasiru Galadima.

**Writing – original draft:** Abubakar Nasiru Galadima, Nor Afiah Mohd Zulkefli, Norliza Ahmad.

**Writing – review & editing:** Abubakar Nasiru Galadima, Norliza Ahmad, Saleh Ngaski Garba.

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
