## [Decision Letter · Decision Letter 0]

26 Aug 2021

PONE-D-21-11468

Theory-Based Immunization Health Education Intervention in Improving Child Immunization Uptake Among Antenatal Mothers Attending Federal Medical Center In Nigeria: A Study Protocol for a Randomized Controlled Trial.

PLOS ONE

Dear Dr. Mohd Zulkefli,

Thank you for submitting your manuscript to PLOS ONE. After careful consideration, we feel that it has merit but does not fully meet PLOS ONE’s publication criteria as it currently stands. Therefore, we invite you to submit a revised version of the manuscript that addresses the points raised during the review process.

We look forward to receiving your revised manuscript.

Kind regards,

Stephen R. Walsh, MDCM

Academic Editor

PLOS ONE

Journal Requirements:

2. PLOS ONE does not copy edit accepted manuscripts (https://journals.plos.org/plosone/s/criteria-for-publication#loc-5). To that effect, please ensure that your submission is free of typos and grammatical errors.

3. Please ensure that you refer to Figure 1 in your text as, if accepted, production will need this reference to link the reader to the figure

Additional Editor Comments (if provided):

Thank you for submitting your protocol to PLoS One. Our reviewers had a large number of concerns with the submission, but are ultimately supportive of the clinical trial proposed as it answers an important medical need. If you believe you can respond adequately to the concerns addressed, we invite you to resubmit the protocol.

Reviewers' comments:

Reviewer's Responses to Questions

**Comments to the Author**

1. Does the manuscript provide a valid rationale for the proposed study, with clearly identified and justified research questions?

Reviewer #1: Partly

Reviewer #2: Yes

Reviewer #3: No

2. Is the protocol technically sound and planned in a manner that will lead to a meaningful outcome and allow testing the stated hypotheses?

Reviewer #1: Partly

Reviewer #2: Yes

Reviewer #3: Yes

3. Is the methodology feasible and described in sufficient detail to allow the work to be replicable?

Reviewer #1: No

Reviewer #2: Yes

Reviewer #3: Yes

4. Have the authors described where all data underlying the findings will be made available when the study is complete?

Reviewer #1: Yes

Reviewer #2: Yes

Reviewer #3: Yes

5. Is the manuscript presented in an intelligible fashion and written in standard English?

Reviewer #1: Yes

Reviewer #2: Yes

Reviewer #3: No

6. Review Comments to the Author

You may also provide optional suggestions and comments to authors that they might find helpful in planning their study.

Reviewer #1: The authors plan an important study. The manuscript submitted requires revision to fully illustrate the rationale for selecting the study design and the particular intervention as well as how it differs from the standard of care. Overall, the manuscript would benefit from more concise presentation of more detailed information, most especially about the methodology planned.

Specifically:

1/Background -

a/Too lengthy; reduce by about 1/3 in length.

b/Focus of the narrative drifts; concentrate on the immunization performance in the selected state, the reasons behind that, how the proposed intervention addresses the gaps and then state the research questions posed.

2/Study design -

a/Explain how this design is the most appropriate to address the research questions.

b/Blinding - It is not possible to blind the participants to the assigned study group because they will know if they participate in the group intervention. Please rectify the description of blinding.

c/Contamination; explain how women who have receive the intervention will be stopped from influencing immunization seeking behavior of the control women.

3/Intervention -

a/Explain the development of the intervention and if it has been tested previously to convince the reader that it is appropriate and potentially feasible. One concern I have is the duration of the single session; in my experience, women in late pregnancy often are not comfortable to spend the whole day at the clinic and should be resting for some of the day.

b/Itemise the components of the intervention. State whether the transport reimbursement and Pampers are considered elements of the intervention.

c/Describe the standard of care in detail. Mention whether women assigned to the control arm will receive similar reimbursements and incentives as those in the intervention arm.

e/Pilot study - There is no explanation of the elements that will be assessed during the pilot stage and whether the women will also be in late pregnancy (35wks).

4/Sample size - The study is powered to detect an effect size of about 20%. Explain whether this difference would lead to meaningful impact in the state immunization program once scaled up, if efficacious.

Reviewer #2: The manuscript entitled ‘Theory-Based Immunization Health Education Intervention in Improving Child Immunization Uptake Among Antenatal Mothers Attending Federal Medical Center In

Nigeria: A Study Protocol for a Randomized Controlled Trial.

Comments

Study Design

The content in study Design (Page 9) and randomization (Page 10) is repeated.

Random selection

Page 12, the information on who prepared and provided the sampling frame list and who performed the recruitment based on the table random numbers to be stated.

Questionnaire quality control

Page 17, the statement ‘The questionnaire is first developed in English language which was later translated into Hausa language via the process of translation and adaptation of instruments figured by the World Health Organization [68]’ is repeated again in the Questionnaire Development section.

Data Analyses

Proper citation for the statistical software to be used.

Descriptive Statistics

How sure mean and sd will be employed in the study and what happened if the data are skewed.

Inferential statistics

1 or 2-tailed test for Fisher’s exact test and Independent t test to be stated. Why independent t test (assuming data are normally distributed) is used when pairwise comparison can be achieved in repeated measures.

For the statement ‘The estimated outputs that will be displayed will then be pooled into a single data set, after which the Generalized Linear Mixed Models (GLMM) analysis’ what estimated outputs and single data set refers to be clearly stated.

Repeated measures one-way analysis of variance (ANOVA) to be written as one-way repeated measures ANOVA.

The actual name for the mixed design repeated measures ANOVA from the statistical software to be used.

It is not clear how these statistical tests (one-way repeated measures ANOVA, GLMM, mixed design repeated measures ANOVA will be employed in the analysis as these tests involved repeated measures. This needs to be clearly explained.

What is the reason one-way repeated measures ANOVA is employed when two groups are involved in the study?

Also for the secondary outcomes, there are many outcomes studied. MANOVA approach could be considered e.g. repeated measures MANOVA etc provided the assumptions are fulfilled.

The subtitle normality test, descriptive statistics inferential statistics to be removed. All the information under these subtitles can be placed under Data Analyses or Statistical Analyses section.

Figure 1 requires cosmetic changes i.e arrows

Some sections in the Materials and Methods require reorganization including statistical analyses and the manuscript requires grammar check and English editing.

Reviewer #3: Great work requiring proper spelling and language check. An excellent protocol that is indeed likely to bring great improvements in an area in Nigeria where immunization if unacceptably low. It is informative and the authors are committed to sharing data on completion

Several words are repeated of misspelt

7. PLOS authors have the option to publish the peer review history of their article (what does this mean?). If published, this will include your full peer review and any attached files.

Reviewer #1: No

Reviewer #2: No

Reviewer #3: No

---

## [Author Response · Author response to Decision Letter 0]

5 Nov 2021

Reviewer #1: The authors plan an important study. The manuscript submitted requires revision to fully illustrate the rationale for selecting the study design and the particular intervention as well as how it differs from the standard of care. Overall, the manuscript would benefit from more concise presentation of more detailed information, most especially about the methodology planned. 

Specifically: 

1/Background -

a/Too lengthy; reduce by about 1/3 in length. 

Has been addressed (kindly refer to page 3-7)

b/Focus of the narrative drifts; concentrate on the immunization performance in the selected state, the reasons behind that, how the proposed intervention addresses the gaps and then state the research questions posed.

Has been addressed (kindly refer to background section)

2/Study design -

a/Explain how this design is the most appropriate to address the research questions.

Randomized controlled trial (RCT) study design is choosing in order to answer the research questions because it is regarded as the most rigorous and robust research method used for determining whether there is an existence of a real cause-effect relation between an intervention and an outcome in health research. Therefore, as we are planning of delivering health education intervention to antenatal mothers RCT will be the best study design to evaluate the effectiveness of our intervention. Moreover, high-quality evidence can be generated through conducting a randomized controlled trial when assessing the effectiveness and safety of an intervention and clinical practice decisions are based on evidence obtained from well-conducted RCTs when available. The main reason why evidence based on RCTs is regarded to be of the highest quality is that evidence based on observational study design is more prone to bias while in evidence based on RCTs, blinding and random allocation of participant to either intervention or control group reduce the likelihood for occurrence of bias. 

b/Blinding - It is not possible to blind the participants to the assigned study group because they will know if they participate in the group intervention. Please rectify the description of blinding. The participants will be blinded against the participants’ groups. The participants wouldn’t be aware of which intervention they will be receiving until the trial is over. 

c/Contamination; explain how women who have receive the intervention will be stopped from influencing immunization seeking behavior of the control women. Contamination will probably be one of the limitations of our study as the possibility of contamination from information sharing which could occur at home or other meeting places cannot also be excluded. However, mothers living in the same household will not be included in order to minimize possible contaminations. 

3/Intervention -

a/Explain the development of the intervention and if it has been tested previously to convince the reader that it is appropriate and potentially feasible. One concern I have is the duration of the single session; in my experience, women in late pregnancy often are not comfortable to spend the whole day at the clinic and should be resting for some of the day.

For the development of the intervention, kindly refer to “development of the immunization health education intervention modules” under the immunization health education intervention modules section (kindly refer to page 15). The intervention has not been tested previously. However, we are in the process of conducting a piloting test as already planned. 

For the duration, it has been amended. 

b/Itemise the components of the intervention.

Has been attached as a separate document (kindly refer to S8 Table 2: component of the intervention (DOCX).

State whether the transport reimbursement and Pampers are considered elements of the intervention.

Has been addressed. (Kindly refer to line 7 of intervention section page 12) 

c/Describe the standard of care in detail. Mention whether women assigned to the control arm will receive similar reimbursements and incentives as those in the intervention arm.

Has been addressed

For standard care (kindly refer to line 6-10 of page 13) and for reimbursements (Kindly refer to line 5 of page 13 under intervention section).

e/Pilot study - There is no explanation of the elements that will be assessed during the pilot stage and whether the women will also be in late pregnancy (35wks).

Has been addressed. (Kindly refer to line 4-8 of quality control of the module content section page 17).

4/Sample size – The study is powered to detect an effect size of about 20%. Explain whether this difference would lead to meaningful impact in the state immunization program once scaled up, if efficacious. 

Yes, it will hopefully lead to meaningful impact as standard formular was used to obtained the study sample size. (kindly refer to line 1-7 of page 10 under sample size determination section and Refer to Supporting document 2). 

Reviewer #2: The manuscript entitled ‘Theory-Based Immunization Health Education Intervention in Improving Child Immunization Uptake Among Antenatal Mothers Attending Federal Medical Center In

Nigeria: A Study Protocol for a Randomized Controlled Trial.

Comments

Study Design

The content in study Design (Page 9) and randomization (Page 10) is repeated.

Has been amended (kindly refer to page 8 and 9). 

Random selection

Page 12, the information on who prepared and provided the sampling frame list and who performed the recruitment based on the table random numbers to be stated.

The sampling frame list will be prepared and provided by the hospital staff working in the Medical Records Department. Likewise, the recruitment based on the table random numbers will be performed by the hospital nurse from Antenatal Care Department. Both of them will not be participated in any of the study process. (kindly refer to page 10 under random selection section). 

Questionnaire quality control

Page 17, the statement ‘The questionnaire is first developed in English language which was later translated into Hausa language via the process of translation and adaptation of instruments figured by the World Health Organization [68]’ is repeated again in the Questionnaire Development section.

Has been amended. (Kindly refer to page 17-18 under questionnaire development section and questionnaire quality control). 

Data Analyses

Proper citation for the statistical software to be used.

Descriptive Statistics

How sure mean and sd will be employed in the study and what happened if the data are skewed. (Kindly refer to line 4-7 of page 18 under data analyses). 

Inferential statistics

1 or 2-tailed test for Fisher’s exact test and Independent t test to be stated. Why independent t test (assuming data are normally distributed) is used when pairwise comparison can be achieved in repeated measures.

Already stated under inferential statistics section. 

For the statement ‘The estimated outputs that will be displayed will then be pooled into a single data set, after which the Generalized Linear Mixed Models (GLMM) analysis’ what estimated outputs and single data set refers to be clearly stated.

Kindly refer to page 19 under data analyses section. 

Repeated measures one-way analysis of variance (ANOVA) to be written as one-way repeated measures ANOVA.

has been amended. 

The actual name for the mixed design repeated measures ANOVA from the statistical software to be used.

Has been addressed. 

It is not clear how these statistical tests (one-way repeated measures ANOVA, GLMM, mixed design repeated measures ANOVA will be employed in the analysis as these tests involved repeated measures. This needs to be clearly explained.

What is the reason one-way repeated measures ANOVA is employed when two groups are involved in the study?

The reason why one-way repeated measures ANOVA is employed is to determine within-group differences with time for knowledge scores, attitude scores, outcome expectation scores, self-efficacy scores, cultural beliefs scores and assumptions on religious regulations scores.

Also for the secondary outcomes, there are many outcomes studied. MANOVA approach could be considered e.g. repeated measures MANOVA etc provided the assumptions are fulfilled.

The subtitle normality test, descriptive statistics inferential statistics to be removed. All the information under these subtitles can be placed under Data Analyses or Statistical Analyses section.

Has been amended. (Kindly refer to page 18-20 under data analyses section). 

Figure 1 requires cosmetic changes i.e arrows

Has been amended. (Refer to page 15 under study process section). 

Some sections in the Materials and Methods require reorganization including statistical analyses and the manuscript requires grammar check and English editing.

Has been addressed. 

Reviewer #3: Great work requiring proper spelling and language check. An excellent protocol that is indeed likely to bring great improvements in an area in Nigeria where immunization if unacceptably low. It is informative and the authors are committed to sharing data on completion

Several words are repeated of misspelt

Has been addressed. 

Review comments 16th August 2021

A suggestion for title modification

A Study Protocol for a Randomized Controlled Trial to evaluate the effect of Theory-Based Immunization Health Education Among Antenatal Mothers Attending Federal Medical Center In Nigeria on Uptake of Child Immunization 

Has been addressed. (kindly refer to line 1-3 of page 1).

Recommendation

An excellent protocol that is indeed likely to bring great improvements in an area in Nigeria where immunization if unacceptably low. It is informative and the authors are committed to sharing data on completion

The manuscript need some revision mainly of the grammar. Simple spell check will correct many of the errors included places where words are repeated. 

Has been addressed entirely.

---

## [Decision Letter · Decision Letter 1]

9 Dec 2021

PONE-D-21-11468R1Theory-Based Immunization Health Education Intervention in Improving Child Immunization Uptake Among Antenatal Mothers Attending Federal Medical Center In Nigeria: A Study Protocol for a Randomized Controlled Trial.PLOS ONE

Dear Dr. Mohd Zulkefli,

Thank you for submitting your manuscript to PLOS ONE. After careful consideration, we feel that it has merit but does not fully meet PLOS ONE’s publication criteria as it currently stands. Therefore, we invite you to submit a revised version of the manuscript that addresses the points raised during the review process.

We look forward to receiving your revised manuscript.

Kind regards,

Stephen R. Walsh, MDCM

Academic Editor

PLOS ONE

Journal Requirements:

Additional Editor Comments (if provided):

Thank you for your resubmission. The reviewers believe that the manuscript has been improved during the revision process but there remain three points to address.

1) The fluency of the written English varies throughout the manuscript. As PLoS ONE does not have any copy-editing services, this will need to be improved before the manuscript can be considered acceptable.

2) The question of "blinding" which Reviewer 1 raised is not a question of labelling. Yes, it is clear that assignment to intervention vs control can be hidden from the study team members who evaluate the outcomes. But it appears from the description of the intervention that it is so different from the control (local standard of care), that it will be obvious to the participants which group they are in. If the participants cannot be blinded (because the intervention is obviously different from the control) then keeping the evaluating team blinded would yield a single-blind study. This could be acceptable but should be described.

3) Reviewer 1 also raised a potential confounding factor. As it appears from the description that participants in the intervention arm will be paid a lot more money than participants in the control arm, it possible that any differences in outcomes could merely be due to the financial incentives. Is there a way to pay the two groups of participants the same amount of money?

Reviewers' comments:

Reviewer's Responses to Questions

**Comments to the Author**

1. Does the manuscript provide a valid rationale for the proposed study, with clearly identified and justified research questions?

Reviewer #2: Yes

Reviewer #3: Yes

2. Is the protocol technically sound and planned in a manner that will lead to a meaningful outcome and allow testing the stated hypotheses?

Reviewer #2: Yes

Reviewer #3: Yes

3. Is the methodology feasible and described in sufficient detail to allow the work to be replicable?

Reviewer #2: Yes

Reviewer #3: Yes

4. Have the authors described where all data underlying the findings will be made available when the study is complete?

Reviewer #2: Yes

Reviewer #3: Yes

5. Is the manuscript presented in an intelligible fashion and written in standard English?

Reviewer #2: Yes

Reviewer #3: Yes

6. Review Comments to the Author

You may also provide optional suggestions and comments to authors that they might find helpful in planning their study.

Reviewer #2: Ensure the naming of supplementary documents in the text and file names are consistent throughout the manuscript.

e.g. (Refer to Supporting document 1), Refer to Supporting document 2, (Refer to supporting document 3), (Refer to Supporting document4), (Refer to Supporting document 5 and 6), Refer to Supporting document 7, (Refer to Supporting document 8), withPan African Clinical Trial Registry PACTR202006722055635(See appendix). registry PACTR202006722055635 (see appendix).

S1 Operational definitions (DOCX). S2 Sample size determination (DOCX). S3 SPIRIT check list (DOCX). S4 Questionnaire (DOCX). S5 Study pro forma (DOCX). S6 Ethical approvals (DOCX). S7 Participants consent form (DOCX). S8 Table 2: component of the intervention (DOCX).

Reviewer #3: This is an important protocol addressing a very pertinent challenge of low vaccine uptake. The author has addressed concerns indicated by the different reviewers.

7. PLOS authors have the option to publish the peer review history of their article (what does this mean?). If published, this will include your full peer review and any attached files.

Reviewer #2: No

Reviewer #3: No

---

## [Author Response · Author response to Decision Letter 1]

6 Jan 2022

1) The fluency of the written English varies throughout the manuscript. As PLoS ONE does not have any copy-editing services, this will need to be improved before the manuscript can be considered acceptable. Has been addressed all over the manuscript. 

2) The question of "blinding" which Reviewer 1 raised is not a question of labelling. Yes, it is clear that assignment to intervention vs control can be hidden from the study team members who evaluate the outcomes. But it appears from the description of the intervention that it is so different from the control (local standard of care), that it will be obvious to the participants which group they are in. If the participants cannot be blinded (because the intervention is obviously different from the control) then keeping the evaluating team blinded would yield a single-blind study. This could be acceptable but should be described. Kindly refer to page 12 line 1-2 under blinding section.

3) Reviewer 1 also raised a potential confounding factor. As it appears from the description that participants in the intervention arm will be paid a lot more money than participants in the control arm, it possible that any differences in outcomes could merely be due to the financial incentives. Is there a way to pay the two groups of participants the same amount of money? Kindly refer to page 13 line 4-7 under intervention section.

---

## [Editor Report · Decision Letter 2]

20 Jan 2022

Theory-Based Immunisation Health Education Intervention in Improving Child Immunisation Uptake Among Antenatal Mothers Attending Federal Medical Center In Nigeria: A Study Protocol for a Randomized Controlled Trial.

PONE-D-21-11468R2

Dear Dr. Mohd Zulkefli,

We’re pleased to inform you that your manuscript has been judged scientifically suitable for publication and will be formally accepted for publication once it meets all outstanding technical requirements.

Kind regards,

Stephen R. Walsh, MDCM

Academic Editor

PLOS ONE

---

## [Editor Report · Acceptance letter]

11 Sep 2022

PONE-D-21-11468R2 

Theory-Based Immunisation Health Education Intervention in Improving Child Immunisation Uptake Among Antenatal Mothers Attending Federal Medical Centre in Nigeria: A Study Protocol for a Randomized Controlled Trial 

Dear Dr. Mohd Zulkefli:

I'm pleased to inform you that your manuscript has been deemed suitable for publication in PLOS ONE. Congratulations! Your manuscript is now with our production department. 

Kind regards, 

on behalf of

Dr. Stephen R. Walsh 

Academic Editor

PLOS ONE